# Resveratrol and Pterostilbene Inhibit SARS-CoV-2 Replication in Air–Liquid Interface Cultured Human Primary Bronchial Epithelial Cells

**DOI:** 10.3390/v13071335

**Published:** 2021-07-10

**Authors:** Bram M. ter Ellen, Nilima Dinesh Kumar, Ellen M. Bouma, Berit Troost, Denise P.I. van de Pol, Heidi H. van der Ende-Metselaar, Leonie Apperloo, Djoke van Gosliga, Maarten van den Berge, Martijn C. Nawijn, Peter H.J. van der Voort, Jill Moser, Izabela A. Rodenhuis-Zybert, Jolanda M. Smit

**Affiliations:** 1Department of Medical Microbiology and Infection Prevention, University Medical Center Groningen, University of Groningen, 9700 RB Groningen, The Netherlands; b.m.ter.ellen@umcg.nl (B.M.t.E.); n.dinesh.kumar@umcg.nl (N.D.K.); e.m.bouma@umcg.nl (E.M.B.); b.h.troost@umcg.nl (B.T.); d.p.i.van.de.pol@umcg.nl (D.P.I.v.d.P.); h.h.metselaar@umcg.nl (H.H.v.d.E.-M.); i.a.rodenhuis-zybert@umcg.nl (I.A.R.-Z.); 2Department of Biomedical Sciences of Cells & Systems, University Medical Center Groningen, University of Groningen, 9700 RB Groningen, The Netherlands; 3Department of Pathology and Medical Biology, GRIAC Research Institute, University Medical Center Groningen, University of Groningen, 9700 RB Groningen, The Netherlands; l.apperloo@umcg.nl (L.A.); m.c.nawijn@umcg.nl (M.C.N.); 4Department of Pediatrics, Beatrix Children’s Hospital, GRIAC Research Institute, University Medical Center Groningen, University of Groningen, 9700 RB Groningen, The Netherlands; d.van.gosliga@umcg.nl; 5Department of Pulmonary Diseases, GRIAC Research Institute, University Medical Center Groningen, University of Groningen, 9700 RB Groningen, The Netherlands; m.van.den.berge@umcg.nl; 6Department of Critical Care, University Medical Center Groningen, University of Groningen, 9700 RB Groningen, The Netherlands; p.h.j.van.der.voort@umcg.nl (P.H.J.v.d.V.); j.moser@umcg.nl (J.M.)

**Keywords:** resveratrol, pterostilbene, SARS-CoV-2, antiviral, human primary bronchial epithelial cells

## Abstract

The current COVID-19 pandemic is caused by severe acute respiratory syndrome coronavirus 2 (SARS-CoV-2) and has an enormous impact on human health and economy. In search for therapeutic options, researchers have proposed resveratrol, a food supplement with known antiviral, anti-inflammatory, and antioxidant properties as an advantageous antiviral therapy for SARS-CoV-2 infection. Here, we provide evidence that both resveratrol and its metabolically more stable structural analog, pterostilbene, exhibit potent antiviral properties against SARS-CoV-2 in vitro. First, we show that resveratrol and pterostilbene antiviral activity in African green monkey kidney cells. Both compounds actively inhibit virus replication within infected cells as reduced virus progeny production was observed when the compound was added at post-inoculation conditions. Without replenishment of the compound, antiviral activity was observed up to roughly five rounds of replication, demonstrating the long-lasting effect of these compounds. Second, as the upper respiratory tract represents the initial site of SARS-CoV-2 replication, we also assessed antiviral activity in air–liquid interface (ALI) cultured human primary bronchial epithelial cells, isolated from healthy volunteers. Resveratrol and pterostilbene showed a strong antiviral effect in these cells up to 48 h post-infection. Collectively, our data indicate that resveratrol and pterostilbene are promising antiviral compounds to inhibit SARS-CoV-2 infection. Because these results represent laboratory findings in cells, we advocate evaluation of these compounds in clinical trials before statements are made whether these drugs are advantageous for COVID-19 treatment.

## 1. Introduction

Within less than a year, the novel severe acute respiratory syndrome coronavirus 2 (SARS-CoV-2) has infected millions of people around the world thereby having a major effect on human health and economy [1]. Infection with SARS-CoV-2 leads to a wide range of manifestations ranging from an asymptomatic infection to a self-limiting mild disease to a potentially fatal disease. It is estimated that approximately 15% of the infected individuals develop severe pneumonia and ~5% develop an acute respiratory distress syndrome (ARDS), septic shock, and/or multiple organ failure. Severe disease is a consequence of lung inflammation and damage caused by direct viral infection of the lung and/or by the immune response triggered to control virus dissemination [2,3,4]. To prevent and treat SARS-CoV-2 infection, vaccines, and antiviral drugs are urgently needed.

The natural compounds resveratrol (trans-3,5,4′-trihydroxystilbene) and the structurally related pterostilbene (trans-3,5-dimethyoxy-4′-hydroxystilbene) share multiple bioactivities which are potentially beneficial for human health [5,6]. In recent years, resveratrol was described to exhibit antiviral activity towards a large number of viruses including human immunodeficiency virus [7], influenza virus [8], respiratory syncytium virus [9], as well as Middle East respiratory syndrome coronavirus (MERS-CoV) [10]. While this work was in progress, resveratrol was also shown to have antiviral activity towards SARS-CoV-2 in Vero E6 cells [11,12]. In most cases, resveratrol was found to directly interfere with viral replication [13]. Next to the direct effect on virus replication, resveratrol was also reported to exhibit anti-inflammatory and antioxidant properties, thereby having the potential to mitigate virus-induced disease pathogenesis [13,14]. Based on these findings, we and others have postulated that resveratrol and its structural analog might be an advantageous treatment option for SARS-CoV-2 infected individuals [15,16,17].

In this study, we examined the efficacy of resveratrol and pterostilbene on suppression of viral replication in several in vitro models of SARS-CoV-2 infection. Specifically, we used African green monkey kidney cells (Vero E6) to delineate the mode of action of resveratrol and pterostilbene by incubating both compounds prior to, during, and after virus cell entry had been established. Furthermore, we determined the number of infected cells and intracellular RNA levels during the course of infection in the absence and presence of compounds. Furthermore, the antiviral effect was evaluated over time to assess the long-lasting effect of the compounds. In addition, we confirmed our findings in a relevant ex vivo model of SARS-CoV-2 infection by assessing the antiviral efficacy in air–liquid interface (ALI) cultured human primary bronchial epithelial cells (PBECs).

## 2. Material and Methods

### 2.1. Compounds

Trans-resveratrol 99% was obtained from Bulkpowders (Colchester, UK) and dissolved in absolute ethanol (EtOH) obtaining a stock solution of 100 mM which was used in the experiments. Pterostilbene 97% HPLC determined (Sigma Aldrich) was dissolved in absolute EtOH obtaining a stock solution of 10 mM which was used in the experiments. The EtOH concentration was below 0.6% in all infection experimental conditions.

### 2.2. Cell Culture and Differentiation

The African green monkey Vero E6 cell line (ATCC CRL-1586) was maintained in Dulbecco’s minimal essential medium (DMEM) (Gibco, Waltham, MA, USA), high glucose supplemented with 10% fetal bovine serum (FBS) (*v/v*) (Life Science Production, Bedford, UK), 1% penicillin (100 U/mL), and 1% streptomycin (100 U/mL) (*v/v*) (Gibco, Waltham, MA, USA). The human lung epithelial cell line Calu-3 (ATCC HTB-55) was maintained in DMEM F-12 (Lonza, Basel, Switserland) supplemented with 10% FBS (*v/v*), 1% Glutamax (*v/v*) (Thermofisher, Waltham, MA, USA), 1% non-essential amino acid (*v/v*) (Thermofisher, Waltham, MA, USA), 1% penicillin (100 U/mL), and 1% streptomycin (100 U/mL) (*v/v*). All cells were mycoplasma negative and maintained at 37 °C under 5% CO_2_. PBECs were cultured from bronchial bushing obtained by fibreoptic bronchoscopy performed using a standardized protocol during conscious sedation [18,19]. The medical ethics committee of the University Medical Center Groningen approved the study, and all subjects gave their written informed consent. The donors were 2 male and 2 female non-smoking healthy control volunteers (less than 2.5 packyears) with no history of respiratory disease aged 49–62. PBECs were cultured and fully differentiated under ALI conditions in transwell inserts, as previously described [20].

### 2.3. SARS-CoV-2 Production and Characterization

The SARS-CoV-2 strain NL/2020 was obtained from European Virus Archive global (EVAg-010V-03903). The original stock was passaged twice in Vero E6 cells to obtain a working stock. Infectious virus titers were determined by plaque assay on Vero E6 cells and defined as the number of plaque-forming units (PFU) per mL. Briefly, Vero E6 cells were seeded at a density of 1.3 × 10^5^ cells/well in a 12-well plate format. At 24 h post-seeding, cells were infected with 10-fold serial dilutions of the sample in duplo. At 2 h post-inoculation (hpi), wells were overlaid with 1% seaplaque agarose (Lonza, Basel, Switserland) prepared in 2× MEM. Plaques were counted at 44 hpi. One plaque in the lowest dilution corresponds to 150 PFU/mL and was set as the detection limit of the assay.

### 2.4. Cytotoxicity Assays

#### 2.4.1. MTS

In the context of Vero E6 and Calu-3 cells, cells were exposed to increasing concentrations of resveratrol and pterostilbene for 8 or 65 h, respectively. Subsequently, cellular cytotoxicity was evaluated using the CellTiter 96^®^ AQueous One Solution Cell Proliferation Assay kit using manufacturer’s instructions from Promega. Briefly, cells were seeded in a 96-well plate at a density of 1 × 10^4^ cells/well. At 24 h post-seeding, Vero E6 cells were treated with increasing concentrations of resveratrol and pterostilbene ranging from 2 to 250 µM or the equivalent volumes of EtOH for 65 h at 37 °C. Calu-3 cells were treated with 50 to 150 µM resveratrol, 40 or 60 µM pterostilbene, or the equivalent volumes of EtOH for 8 h at 37 °C. At 8 h post-treatment, 20 µL of MTS/PMS solution was added per well and incubated for 2 h at 37 °C. Subsequently, 10% SDS (*v/v*) was added to each well (2% end concentration) to stop the reaction and the absorbance was measured at 490 nm with a microplate reader. Values are displayed as percentage compared to normalized non-treated (NT) control. All individual experiments were performed in triplicate.

#### 2.4.2. Live Death Staining Flow Cytometry

PBECs were exposed to 150 µM resveratrol and 60 µM pterostilbene at the basolateral side for 48 h at 37 °C, and subsequently harvested and stained with fixable viability dye eFluor 780 for 20 min at 4 °C. After staining cells were washed in FACS buffer (PBS 2% FBS (*v/v*)), 5 mM EDTA), centrifuged and subsequently fixed with 4% PFA for 10 min at 4 °C. After fixation, cells were washed, centrifuged, and resuspended in FACS buffer. Cells were analyzed for viability with the LSR-2 flow cytometer (BD Bioscience, Franklin Lakes, NJ, USA). Data were analyzed using Kaluza software (Beckman Coulter, Brea, CA, USA).

#### 2.4.3. LDH

PBECs were exposed to 150 µM resveratrol and 60 µM pterostilbene at the basolateral side for 48 h at 37 °C. After incubation, apical sides of the inserts were incubated with medium for 30 min at 37 °C. The apical wash was collected and centrifuged 2000× *g* at 4 °C to clear from cell debris. The commercially available kit (CyQUANT™ LDH Cytotoxicity Assay Kit, Thermofisher, Waltham, MA, USA) was used according to manufacturer protocol. The absorbance was measures at 490 and 680 nm using a microplate reader; 680 nm absorbance OD values (background) were subtracted from 490 nm OD values. Cytotoxicity was calculated using the following formula:(1)$$\%\;cytotoxicity=\frac{\left(OD\;Compound\;treated\;LDH\;activity-OD\;spontaneous\;LDH\;activity\right)\;}{\left(OD\;maximum\;LDH\;activity-OD\;spontaneous\;activity\right)}$$

### 2.5. Antiviral Assay in Vero E6 and Calu-3 Cells

#### 2.5.1. Infectivity Assay

Vero E6 cells were seeded at a density of 1.3 × 10^5^ cells/well in 12-well plates and Calu-3 cells were seeded at a density of 2 × 10^5^ cells/well in 24-well plates. The following day, cells were infected with SARS-CoV-2 at a multiplicity of infection (MOI) 1 and treated with increasing concentrations of resveratrol and pterostilbene or the equivalent volumes of EtOH corresponding to the highest concentration of compound for 2 h at 37 °C. Infection was done in 250 µL DMEM 2% FBS (*v/v*) medium. After infection, virus inoculum was removed, cells were washed twice with plain DMEM media, and fresh DMEM 10% FBS (*v/v*) containing the compound or the equivalent volumes of EtOH was added after which incubation was continued. Cell supernatant was collected at 8 hpi, centrifuged to clarify from cell debris, and the viral titer was determined using plaque assay. For the durability assay, Vero E6 cells were infected with SARS-CoV-2 at MOI 0.01 and treated with 150 µM resveratrol or 60 µM pterostilbene or the equivalent volume of EtOH as indicated above. Supernatants were collected at 16, 24, 40, and 60 hpi and analyzed as above.

#### 2.5.2. NSP8 Detection Flow Cytometry

Vero E6 cells were seeded and infected as described above for the infectivity assay. At 8 hpi, cells were harvested and fixated with 4% PFA for 10 min at 4 °C. Cells were washed, centrifuged, and subsequently stained with mouse anti-SARS-CoV-2 NSP8 monoclonal antibody (Cat no: GTX632696, GeneTex, Irvine, CA, USA) at a 1:1000 dilution in FACS buffer for 30 min at 4 °C. Upon incubation, cells were again washed, centrifuged, and incubated with rabbit anti-mouse- Alexa647 (Thermofisher, Waltham, MA, USA) at a 1:1000 dilution for 30 min in the dark at 4 °C. Upon incubation, cells were washed, centrifuged, and resuspended in FACS buffer. Cells were analyzed with the NovoCyte Quanteon Flow Cytometer (Agilent Technologies, Santa Clara, CA, USA). Data were analyzed using Kaluza software (Beckman Coulter, Brea, CA, USA).

#### 2.5.3. qPCR RdRp Gene

Vero E6 cells were seeded and infected as described above for the infectivity assay. At the indicated times post-infection, cells were harvested and lysed with 350 µL RLT buffer 1% β-ME. Viral RNA was isolated using the RNeasy mini kit (Qiagen, Hilden, Germany) according to manufacturer’s protocol. cDNA synthesis from viral RNA was performed using Omniscript RT kit (Qiagen, Hilden, Germany) with the reverse primer CARATGTTAAASACACTATTAGCATA. Next, qPCR was performed by using the Qiagen Hot star taq polymerase kit in combination with the forward primer GTGARATGGTCATGTGTGGCG, reverse primer CARATGTTAAASACACTATTAGCATA, and RdRp_SARSr-P2 (5′FAM/3′BHQ) probe CAGGTGGAACCT CATCAGGAGATGC.

DNA amplification was done at 50 °C 120 s, 95 °C 900 s, and subsequently 40 cycles of 95 °C 15 s, 60 °C 60 s.

### 2.6. Antiviral Assay in Primary Bronchial Epithelial Cellsd

After 3 weeks culture under ALI conditions, cells were washed once with 1:1 plain DMEM and Airway Epithelial Cell Growth medium (Promocell, Heidelberg, Germany) supplemented with BSA and supplement kit (ALI culture medium, Promocell, Heidelberg, Germany), and inoculated with SARS-CoV-2 at MOI 5 at the apical side. At the time of infection, 75 or 150 µM resveratrol, 60 µM pterostilbene or the equivalent volumes of EtOH were added at the basolateral side of the insert in ALI culture medium. At 2 hpi, cells were washed twice with OptiMEM (Gibco, Waltham, MA, USA) at the apical side and incubation was continued on air at 37 °C. Thirty min prior to harvesting (12, 24, and 48 hpi), OptiMEM was added to the apical side of the ALI cultures. At the time of harvest, the apical supernatant and 150 µL of basolateral medium was harvested. At the basolateral side, after each harvest new ALI culture media containing the compound or EtOH was added. The viral titer in the apical supernatant was determined using plaque assay.

### 2.7. Time-of-Drug-Addition Assay

For the time-of-drug-addition experiments, the cells were treated with 150 µM resveratrol or 60 µM pterostilbene at pre-, during, or post-inoculation conditions. For the pre-inoculation condition, cells were incubated with the compounds or the equivalent volume of EtOH for 2 h. At the time of infection, cells were washed three times before the addition of the virus inoculum. For the during condition, the compounds or the equivalent volumes EtOH were added together with the virus inoculum and was present for 2 h. At 2 hpi, cells were washed three times with plain DMEM, fresh DMEM 10% FBS was added, and incubation was continued. For the post-inoculation conditions, the compounds or the equivalent volumes EtOH were added to the cell culture medium after removal of the virus inoculum. All supernatants were collected 8 hpi, centrifuged to clarify from cell debris and subjected to plaque assay to determine the viral titer.

### 2.8. Virucidal Assay

2.5 × 10^5^ PFU of SARS-CoV-2 were incubated in 300 µL DMEM 2% FBS (*v/v*) in the presence or absence of 150 µM resveratrol, 60 µM pterostilbene or the equivalent volume of EtOH for 2 h at 37 °C. Subsequently, viral titer was determined by plaque assay.

### 2.9. Statistical Analysis

All data are represented as mean ± SEM. The concentration that reduced virus particle production by 50 and 90% is referred to as EC50 and EC90, respectively. Dose–response curves were fitted by nonlinear regression analysis employing a sigmoidal model. All data was analyzed in GraphPad Prism 8 software. Non-paired two-tailed Student T test was used to evaluate statistical differences and a *p* value ≤ 0.05 was considered significant with * *p* ≤ 0.05, ** *p* ≤ 0.01, and *** *p* ≤ 0.001 and NS as non-significant.

## 3. Results

### 3.1. Resveratrol and Pterostilbene Inhibit SARS-CoV-2 Infection in Vero E6 Cells

Prior to assessing the antiviral activity of resveratrol and pterostilbene, we determined the cellular cytotoxicity of the compounds in Vero E6 cells. We observed a dose-dependent cytotoxic effect of both resveratrol and pterostilbene (Figure A1A,B). Limited cytotoxicity was observed up to a concentration of 200 µM resveratrol and 100 µM pterostilbene. At these conditions, no cytotoxicity was observed for the solvent control EtOH. However, the cells treated with compound appeared to be stressed based on morphology, and therefore we decided to use 150 µM resveratrol and 60 µM pterostilbene as the highest concentration in follow-up experiments. Next, we investigated the antiviral effect of resveratrol and pterostilbene during infection with SARS-CoV-2 (isolate NL/2020). Vero E6 cells were inoculated with SARS-CoV-2 (MOI 1) in the presence of increasing concentrations of resveratrol, pterostilbene, or an equivalent volume of EtOH corresponding to the highest concentration of the compound. Resveratrol and pterostilbene showed a dose-dependent antiviral effect on SARS-CoV-2 infection in Vero E6 cells (Figure 1A,B). No effect on infectious virus production was observed for the EtOH solvent control. Subsequent nonlinear regression analyses revealed that virus particle production is reduced by 50% (EC50) at a concentration of 66 µM resveratrol and 19 µM pterostilbene (Figure 1C). Furthermore, a 90% reduction (EC90) of virus progeny is observed at a concentration of 119 µM resveratrol and 47 µM pterostilbene (Figure 1C). Thus, pterostilbene has a more efficacious antiviral effect at lower concentrations in comparison to resveratrol. While our work was in progress, other studies [11,12] revealed an EC50 value of approximately 5 µM for resveratrol, yet in these studies the EC50 value was based on a reduction in RNA levels and assessed at different experimental conditions such as a lower MOI (Yang et al. used an MOI of 0.01 for infection) and determination of the effect at 48 hpi. Therefore, no direct comparison can be made, yet it is clear that resveratrol has an antiviral effect.

Next, we investigated how long resveratrol and pterostilbene maintain their antiviral activity in cell culture when added at the highest non-toxic concentration. We infected Vero E6 cells with SARS-CoV-2 at MOI 0.01 in the presence of the compounds and harvested the supernatant at 16, 24, 40, and 60 h post-inoculation (hpi). Importantly, no cytotoxic effects were seen upon exposure of the compounds to the cells for 65 h (Figure A1C,D). For the non-treated (NT) and solvent control samples, no differences were observed in virus growth and virus particle production plateaued at 40 hpi at which time point all cells were dead (Figure 1D,E). In the presence of resveratrol as well as pterostilbene a strong antiviral effect (~2 log reduction which corresponds to 99% reduction in virus production) was observed at 16 and 24 hpi. Significant antiviral activity was observed up to 40 hpi. Thus, without replenishment of the compound an antiviral effect is seen up to 5 rounds of replication, as 1 round of replication is approximately 8 h [21]. These findings highlight the long-lasting in vitro antiviral effect of both compounds.

### 3.2. Resveratrol and Pterostilbene Interfere with SARS-CoV-2 Infectivity

To test whether the antiviral capacity of the compounds rely on lysis, inactivation, or neutralization of the virion itself, we next performed a virucidal assay [22]. Briefly, 2.5 × 10^5^ PFU of SARS-CoV-2 was incubated with 150 µM resveratrol, 60 µM pterostilbene, or equivalent volumes of EtOH for 2 h and subjected to plaque assay. No differences in viral titers were observed relative to the NT or EtOH control (Figure 2A). This indicates that the compounds do not exhibit virucidal activity at these conditions, but rather interfere with viral replication in Vero E6 cells. To examine this further, we performed a time-of-drug-addition experiment. In this experiment, resveratrol or pterostilbene was added prior, during, or post SARS-CoV-2 inoculation and infectious virus particle production was evaluated at 8 hpi (Figure 2B). Comparable results were obtained for both compounds (Figure 2C,D). No effect was observed when the compounds were solely present prior to inoculation. A mild, yet non-significant, reduction in virus particle production was observed when the compounds were present during virus inoculation (*p*-value 0.077). Importantly, a significant reduction in virus particle production was observed when the compounds were added after removal of the virus inoculum (Figure 2C,D). This suggests that both compounds predominantly act after virus cell entry and membrane fusion. To further delineate the mode of action we next added the compounds at 4 and 6 hpi and harvested the supernatant at 8 hpi to determine the production of virus progeny. Addition of resveratrol at 4 hpi led to a non-significant reduction in virus progeny (*p*-value 0.076) and addition at 6 hpi had no effect on the secretion of virus progeny. For pterostilbene, no significant antiviral effect was seen when added at 4 and 6 hpi (Figure A2A,B). Next, we investigated the effect of the compounds on the number of infected cells. To this end, we analyzed the expression of non-structural protein 8 (NSP8) at 8 hpi using flow cytometry. In case of resveratrol, the number of infected cells was significantly reduced from 50% in the solvent control to 15% in the resveratrol-treated condition (Figure 2E). In case of pterostilbene, the percentage of infected cells was 58% for the solvent control and 29% for pterostilbene-treated cells. Albeit a negative trend is seen, this difference was found non-significant (*p*-value 0.053). The NSP8 mean fluorescence intensity of the infected cell population was slightly reduced in resveratrol- and pterostilbene-treated cells, suggesting that the addition of resveratrol and pterostilbene to the cells may also influence the expression level of NSP8 within infected cells (Figure 2F). Next, we aimed to assess the effect of the compounds on intracellular viral RdRp RNA copy levels at 2, 4, 6, and 8 hpi using Q-RT-PCR. In control experiments (i.e., non-treated and solvent-treated cells), the Ct values remained constant up to 4 hpi (Ct ~20) and started to decrease at 6 hpi (Ct ~18), suggesting that in this experimental setup active viral RNA replication can only be detected at 6 hpi. At 8 hpi, the Ct value further decreased to 15.5 for non-treated control and on average 16.2 for solvent-treated controls conditions (Figure 2G,H). At 2 and 4 hpi, the Ct value likely reflects incoming viral RNA or low-level RNA replication (i.e., that ongoing degradation of the incoming viral RNA is equal to production of progeny RNA copies). To obtain an idea on the sensitivity of our Q-RT-PCR, we determined the Ct value of a dilution range of our stock virus and revealed that 4 Ct values correspond to a 10-fold difference in PFU titer, indicating that our Q-RT-PCR is in fact quite sensitive (Figure A2D). In resveratrol-treated cells, an average Ct value of 20.6 and 20 was determined at 6 and 8 hpi, respectively (Figure 2G). In pterostilbene-treated cells, an average Ct value of 19.9 was determined at 6 hpi, which decreased to 17.6 at 8 hpi. The Ct value in resveratrol-treated cells was found comparable to those in non-treated conditions at 2 and 4 hpi and suggests that RNA replication is strongly impaired. For pterostilbene, RNA replication is also impaired albeit to a lower extent than resveratrol at 8 hpi. Collectively, given that the most potent antiviral effect is seen when the compounds are added at post-infection conditions (Figure 2C,D), it is tempting to speculate that both resveratrol and pterostilbene actively interfere with RNA replication (Figure 2G,H), thereby lowering the chance for SARS-CoV-2 to productively infect a cell (Figure 2E).

### 3.3. Resveratrol and Pterostilbene Do Not Exhibit Significant Anti-SARS-CoV-2 Activity in Calu-3 Cells

We next sought to verify the findings in the human lung epithelial cell model, Calu-3 cells [23]. Resveratrol was more cytotoxic in Calu-3 cells when compared to Vero E6, as the highest non-toxic concentration was 50 µM (Figure A3A) At this concentration, no antiviral effect was observed in Vero E6 cells. For pterostilbene, the highest non-toxic concentration was set at 60 µM, similar to Vero E6 cells (Figure A3B). After inoculation with SARS-CoV-2 MOI 1, no significant reduction in virus particle production was observed with both compounds present at the highest non-toxic concentrations in Calu-3 cells (Figure 3A,B) although a negative trend in virus particle production was observed with pterostilbene. The inability of both compounds to induce antiviral effects in Calu-3 as observed in Vero E6, could underline the inherent differences between these two cell line models.

### 3.4. Resveratrol and Pterostilbene Significantly Inhibit SARS-CoV-2 Infection in Primary Human Bronchial Epithelial Cells Cultured under ALI Conditions

As SARS-CoV-2 replication is initiated in the upper respiratory tract, we next studied the antiviral effect in human primary epithelial cells (PBEC) differentiated under ALI culture conditions [24,25,26]. The epithelial cells were isolated from the bronchi of healthy volunteers and differentiated at ALI for approximately 3 weeks to yield ciliated and secretory epithelial cells [27] (Figure 4A). In this model, no cytotoxicity was found at a concentration of 150 µM resveratrol and 60 µM pterostilbene as determined by live/death staining using flow cytometry and by a LDH assay (Figure A4A,B). Accordingly, fully differentiated PBECs were inoculated with SARS-CoV-2 (MOI 5) in the presence of resveratrol and pterostilbene, and progeny virus particle production was evaluated at 12, 24, and 48 hpi (Figure 4A). At 12 hpi, an antiviral effect was observed as in presence of the compounds the titer often fell below the threshold of detection (Figure 4B,C). Progeny virus particle production increased over time and at 24 and 48 hpi a significant antiviral effect was observed for both resveratrol and pterostilbene (Figure 4B,C). At 48 hpi, the virus titer was significantly reduced with 2.1 Log (corresponding to 99.3% reduction) when compared to the EtOH control in PBECs treated with resveratrol. In presence of pterostilbene, the virus titer was reduced with 1.2 Log (corresponding to 87.5% reduction) when compared to the EtOH control in PBECs at 48 hpi. Collectively, our data demonstrate that resveratrol and pterostilbene exhibit potent antiviral activity towards SARS-CoV-2 in differentiated human PBECs.

## 4. Discussion

Based on our data, we conclude that both resveratrol and pterostilbene have the potential to exhibit antiviral efficacy early in the course of COVID-19. Resveratrol or pterostilbene strongly reduced SARS-CoV-2 production not only in Vero-E6 cells, but also in the biologically highly relevant primary human bronchial epithelial cell model. Moreover, we show that both compounds actively interfere with the infectious replication cycle of the virus and, without replenishment of the compound, exert antiviral activity up to roughly five replication cycles of SARS-CoV-2 in vitro.

Resveratrol and pterostilbene share many biological functions [28]. Time-of-drug-addition experiments revealed that resveratrol and pterostilbene strongly inhibited SARS-CoV-2 production when administered after removal of the virus inoculum, suggesting that they both act after virus entry into the host cells Furthermore, resveratrol was found to decrease the number of infected cells and drastically reduced the intracellular RNA levels. For pterostilbene a similar trend was observed albeit to a lower extent. Based on the above findings, we hypothesize that both compounds interfere with the early stages of virus replication thereby reducing the chance to productively infect a cell. Alternatively, upon virus cell entry, protein translation is impaired thereby reducing RNA replication and the chance to productively infect a cell. The later hypothesis is not very likely as only a slight reduction in protein expression levels was seen in resveratrol-treated infected cells at late time points. Our observations are also in line with the antiviral activity of resveratrol against other viruses. For example, resveratrol inhibited influenza virus replication and viral protein synthesis via interference with nuclear–cytoplasmic transport of vRNA and the inhibition of protein kinase C activity [8]. Resveratrol also decreased viral gene expression and viral protein synthesis of Herpes Simplex Virus and Epstein–Barr virus [29,30,31]. Future studies should delineate the in-depth mode of action of resveratrol and pterostilbene in controlling SARS-CoV-2 infection.

Next to the direct antiviral properties on SARS-CoV-2 replication, resveratrol might also be suitable to moderate the exacerbated inflammatory response observed in COVID-19 patients [32]. Resveratrol has been shown to have anti-inflammatory properties thereby reducing the release of proinflammatory cytokines [9,33,34,35]. Second, as an antioxidant, resveratrol is found to prevent the formation of reactive oxygen species (ROS), prevent airway epithelial remodeling by upregulation of SIRT1 and activate superoxide dismutase (SOD) [36,37]. The activation of SIRT1 has also been shown to deacetylate HMGB1, resulting in the upregulation of antiviral interferon stimulating genes [38]. It may therefore be possible that the antiviral effect observed at late time points in PBECs may also be due to the onset of innate immunity. Resveratrol was also observed to reduce the secretion of leptin [39], an adipokine that has been implicated to be a contributing factor in the development of respiratory failure and ARDS in SARS-CoV-2 infected patients [17]. Therefore, resveratrol might not only be a promising candidate to inhibit viral replication early in infection it may also alleviate disease symptoms later in infection.

Pterostilbene was found more effective in controlling SARS-CoV-2 replication than resveratrol at low concentrations. Indeed, many in vitro and animal studies have revealed that pterostilbene is superior to resveratrol. This has been linked to the dimethoxy structure in pterostilbene which increases lipophilicity and restricts glucuronidation and sulfation hence improving the solubility, absorption, and bioavailability of the compound [40]. In healthy volunteers, oral administration of both compounds is considered safe at doses up to 5 g per day for resveratrol and 250 mg per day for pterostilbene [41,42]. However, the bioavailability of resveratrol is low as plasma concentrations in the nanomolar range were achieved [43,44]. The pharmacokinetics of pterostilbene has not been investigated in humans and given its favorable characteristics and potent antiviral activity towards SARS-CoV-2 infection this is highly warranted.

Comparable to other respiratory viruses, potent antiviral activity towards SARS-CoV-2 is seen at relatively high concentrations of resveratrol. Due to the low bioavailability of resveratrol upon oral administration, it is unlikely that an effective antiviral dose towards respiratory viruses can be achieved in humans via this route of administration. Therefore, other administration modes and routes have been explored. Promising strategies for antiviral treatment are the use of aerosolized suspension sprays of resveratrol, co-spray dried microparticles or nanotechnology approaches (e.g., nanoparticles and nanosponges) [45,46,47]. Although not yet tested, these strategies might also further enhance the potency of pterostilbene. Adjacent to the improved bioavailability, some of these techniques allow drug directed and local delivery of resveratrol or pterostilbene to the primary site of active SARS-CoV-2 replication using inhalation-based systems, which are currently considered as a favorable future strategy for the treatment of COVID-19 [48].

Finally, as both drugs are commercially available and could therefore be used as a self-medicative prophylactic, a word of caution must be considered. Our data represent promising laboratory findings in cells, and therefore do not indicate that these drugs will be of benefit to treat COVID-19 in patients. Randomized double-blind controlled clinical trials must first swiftly be conducted to prove whether these drugs are indeed advantageous for COVID-19 treatment.

## Figures and Tables

**Figure 1 viruses-13-01335-f001:**
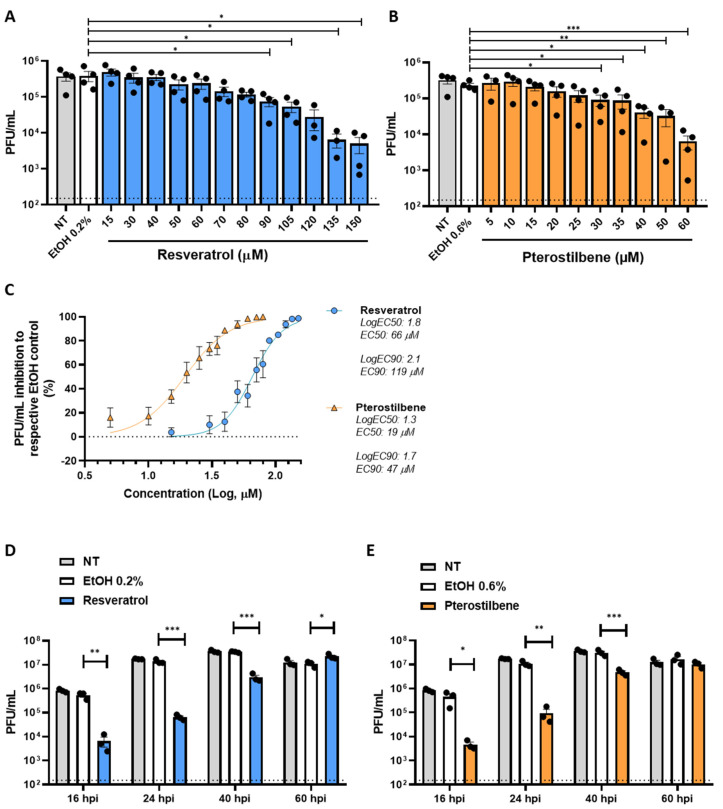
Antiviral effect of resveratrol and pterostilbene towards SARS-CoV-2 in Vero E6 cells. Production of infectious virus particles by Vero E6 cells inoculated with SARS-CoV-2 at MOI 1 in the absence (NT denotes for non-treated) or presence of increasing concentrations of (**A**) resveratrol, (**B**) pterostilbene or (**A**,**B**) the EtOH solvent control. (**C**) The EC50 and EC90 values determined by nonlinear regression analysis. (**D**,**E**) Durability of the antiviral effect of (**D**) resveratrol 150 µM and (**E**) pterostilbene 60 µM at 16, 24, 40, and 60 hpi. Dotted line indicates the threshold of detection. Data are represented as mean ± SEM of at least three independent experiments. Each symbol represents data from a single independent experiment. Student T test was used to evaluate statistical differences and a *p* value ≤ 0.05 was considered significant with * *p* ≤ 0.05, ** *p* ≤ 0.01 and *** *p* ≤ 0.001. In the absence of ‘*’ the data are non-significant.

**Figure 2 viruses-13-01335-f002:**
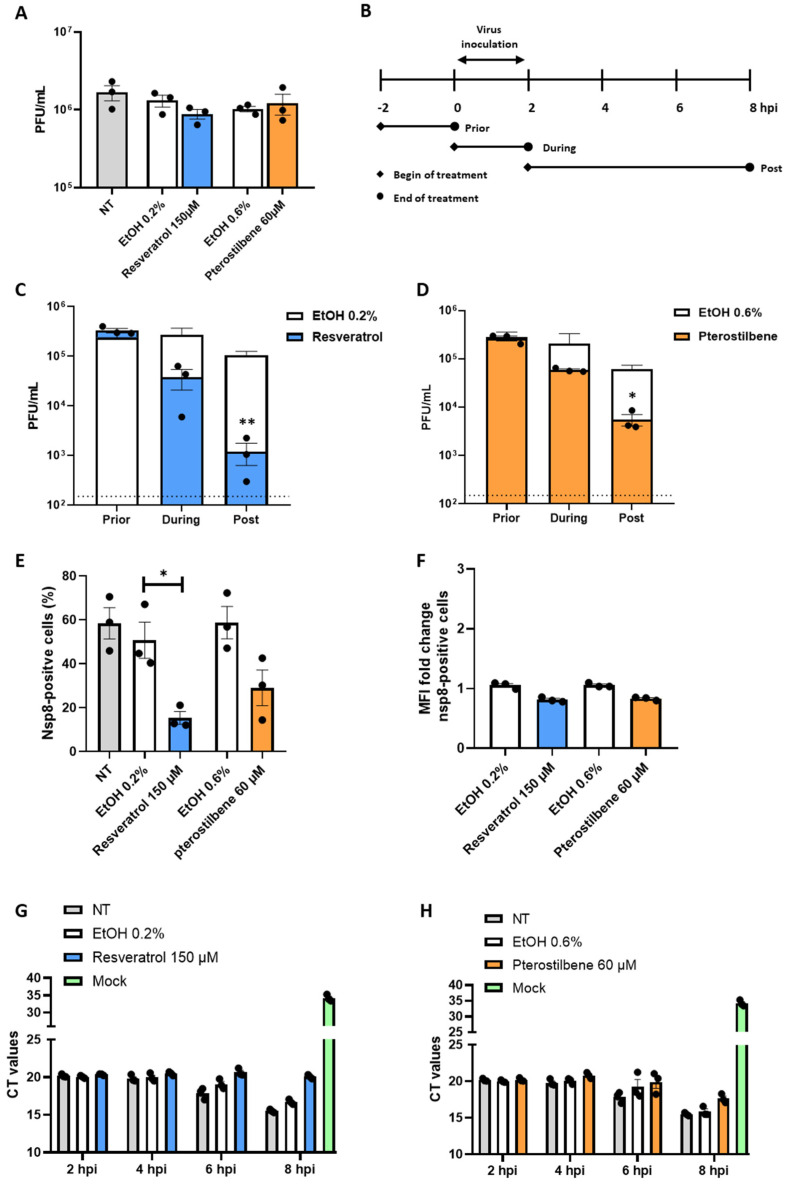
Resveratrol and pterostilbene interfere with SARS-CoV-2 infection when added post virus inoculation conditions. (**A**) Virucidal effect of resveratrol and pterostilbene on SARS-CoV-2. (**B**) Schematic representation of the experimental design of the time-of-drug-addition assay. Vero E6 cells were inoculated with SARS-CoV-2 at MOI 1 and treated with (**C**,**E**,**F**,**G**) 150 µM resveratrol, (**D**,**E**,**F**,**H**) 60 µM pterostilbene or (**C**–**H**) equivalent volumes of EtOH. Virus production was determined at 8 hpi via plaque assay (**C**,**D**). The number of NSP8-positive cells were determined at 8 hpi via flowcytometry (**E**,**F**). Intracellular RNA levels were determined at 2,4,6 and 8 hpi via qPCR (**G**,**H**). Data are represented as mean ± SEM, symbols represent three independent experiments. Dotted line indicates the threshold of detection. Student T test was used to evaluate statistical differences and a *p* value ≤ 0.05 was considered significant with * *p* ≤ 0.05. In the absence of ‘*’ the data are non-significant.

**Figure 3 viruses-13-01335-f003:**
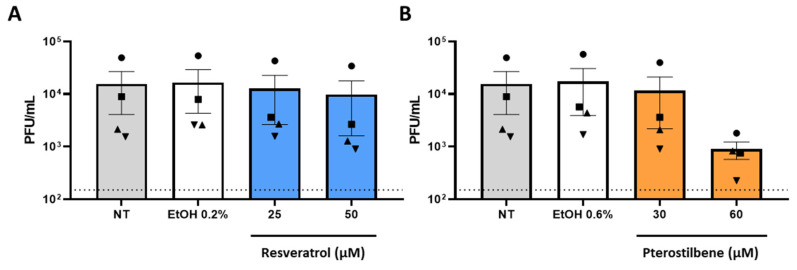
Resveratrol and pterostilbene do not significantly inhibit SARS-CoV-2 infection in Calu-3 cells. Calu-3 cells were inoculated with SARS-CoV-2 MOI 1 in the absence (non-treated, NT) or presence of (**A**) 25 or 50 µM resveratrol, (**B**) 30 or 60 µM pterostilbene, or (**A**,**B**) the equivalent volumes of EtOH. Virus particle production was determined at 8 hpi by plaque assay. Data are represented as mean ± SEM, symbols represent four independent experiments. Dotted line indicates the threshold of detection. Student T test was used to evaluate statistical differences and a *p* value ≤ 0.05 was considered significant. In the absence of ‘*’ the data is non-significant.

**Figure 4 viruses-13-01335-f004:**
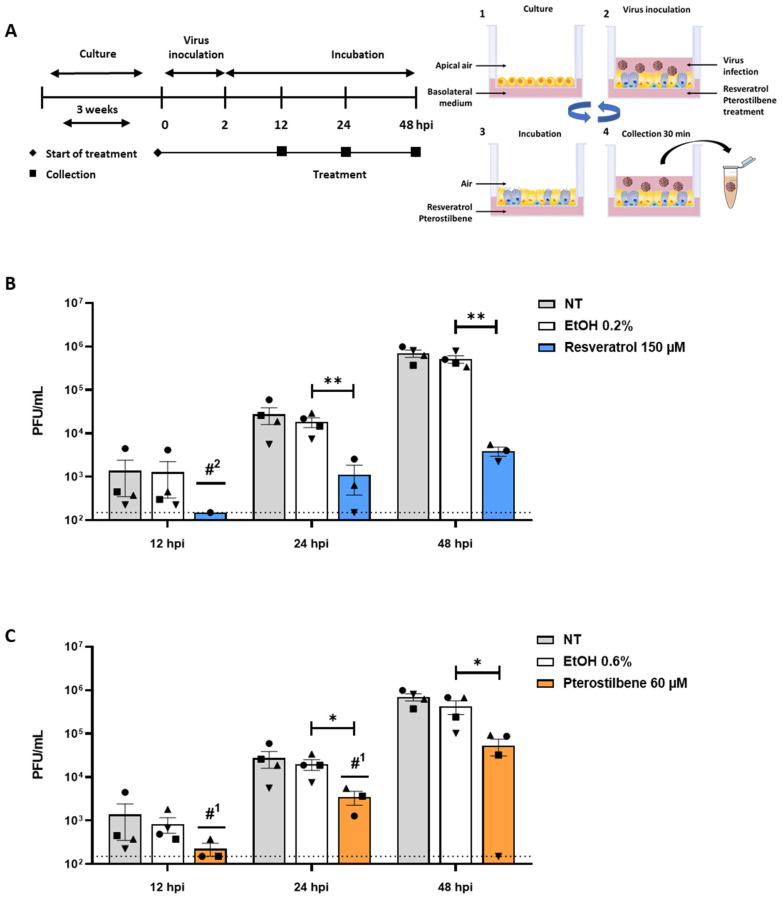
Resveratrol and pterostilbene inhibit SARS-CoV-2 infection in primary human bronchial epithelial cells. (**A**) Schematic representation of the experimental design. (1) Primary human bronchial epithelial cells (PBECs) were cultured on permeable inserts under Air Liquid Interface (ALI) conditions for 21 days. (2) Cells were inoculated with SARS-CoV-2 MOI 5 at the apical side. (3) At 2 hpi, the virus inoculum was removed and cells were exposed to air until virus collection. (4) Medium was added to the apical side and collected after 30 min incubation. After supernatant collection, PBECs were exposed to air again (3) until the next collection time point. Steps 4 and 3 were repeated until the end of the experiment. (**B**,**C**) Virus particle production in PBECs in presence of (**B**) resveratrol or (**C**) pterostilbene. Scheme adapted from STEMCELL Technologies. Data are represented as mean ± SEM, symbols represent three or four different donors. Dotted line indicates the threshold of detection. The number of values below detection limit are indicated with #^X^. Student T test was used to evaluate statistical differences and a *p* value ≤ 0.05 was considered significant with * *p* ≤ 0.05, ** *p* ≤ 0.01. In the absence of ‘*’ the data is non-significant.

## Data Availability

Data is contained within the article.

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
