# Peer review of "Resveratrol and Pterostilbene Inhibit SARS-CoV-2 Replication in Air–Liquid Interface Cultured Human Primary Bronchial Epithelial Cells"

_viruses, 2021, doi:10.3390/v13071335_

Round 1

Reviewer 1 Report

The authors have worked hard to improve this and seem to have taken the criticism offered in the way it was intended.

I’m happy for this to be published in its current form.

Reviewer 2 Report

In this paper, Ellen, Kumar et al characterize the effects of two compounds, Resveratrol and Pterostilbene, in SARS CoV2 infection. They first deeply characterize the effect on SARS-CoV-2 infection in vitro using the canine Vero E6 cell line. Using a combination of assays, they properly showed a potent anti-viral activity of both compounds, at non-toxic doses. Through a drug-addition assay, they showed the efficacy of drugs if administered throughout the infection, while pretreatment of Vero cells is not sufficient to reduce SARS CoV-2 infection, thus suggesting that drugs are targeting post-entry processes. The levels of viral protein expression (NSP8) and vRNA production (qRT-PCR) are both reduced upon compounds treatment, suggesting that the viral replication per se is affected. The authors then explored the effect of both compounds in other cell systems: the human Calu-3 cell line and differentiated primary human bronchial epithelial cells (PBEC). While Calu-3 infection doesn’t vary under Resveratrol or Pterostilbene treatment, the more relevant PBEC system is sensitive to compounds treatment which leads to a reduction of SARS CoV2 infection. Thus, this article clearly indicates that both resveratrol and pterostilbene have antiviral activity against SARS-CoV-2 that would deserve further exploration as antiviral therapeutics.

The reviewer agrees that the results present warrant publication, despite other papers have previously reported the efficacy of the Resveratrol against SARS-CoV-2 infection, which are well referred by the authors. That, for the following reasons: (i) the present work is by far more comprehensive regarding the characterization of resveratrol against SARS-CoV-2 infection, above all owing to the use of cells systems more relevant that Vero E6 cells, especially differentiated PBEC. (ii) The use of two related compounds increases the robustness of the data

This reviewer is therefore clearly in favor of publishing this article, which I think is both concise and clearly written. I would nevertheless suggest several minor modifications to further enhance the quality of the manuscript as follows:

> general comments: the authors present PFU/mL of their biological replicates in all the figures. In Calu3 and PBEC cell systems, the in vitro infection looks like being less efficient than in Vero E6, and more sensitive to experimental variations, which is clearly seen in the presented figures. If the authors represent ratio to non-treated within each of their replicate, then the drug effect would appear independently of the experimental variations. I suggest the authors present it (at elast in a supp figure).

> as mentioned by the authors, anti-viral effects of resveratrol have been reported in previous works. The authors should point to the added values of their present work.The indicated IC50 for resveratrol in Vero E6 is around 60 µM, which doesn’t fit with the reported EC50 of the same drug by others. (± 5µM). The authors should comment on that point.

- line 262 : precise which concentration of compounds is used for low-MOI, long time pi settings

- line 304 : remove “mild” since the reduction is of 1 log that is not so mild while not significant

-line 308 : is it a particular reason for labelling for NSP8 instead of N protein which is the most commonly used marker of SARS-CoV-2 infection?

- line 357 : state the moi used for Calu-3 infection

- line 382 : how differentiation of PBEC has been assessed?

- line 386 : the lack of detection of viral replication at 12 hr pi of treated cells already indicates a reduction of viral replication upon compounds treatment

Reviewer 3 Report

This manuscript by ter Ellen et al. comprises a study evaluating inhibition ability of resveratrol and pterostilbene in SARS-CoV-2 infected human primary bronchial epithelial cells. They used plaque-forming assay and qRT-PCR method to compare mock-treated and chemical treated cell lines including VeroE6, Calu-3, and primary human cells after SARS-CoV-2 infection. In the manuscript, the author showed that there is a significant inhibition effect when using both resveratrol and pterostilbene during the infection with approximately two logs reducing in the virus titer up to 48 hours post-infection. Although the bioavailability of resveratrol upon oral administration was low and cannot reach the same amount as in the cell culture, the authors suggested that these chemicals can be further combined with other techniques to increase its bioavailability to a follow-up application SARS-CoV-2 treatment. 

Major and minor points: 
1)    The use of SARS-CoV-2 MOI: in line 204, the MOI used for ALI culture was up to 5, could the author explain why such a high amount of virus should be used? 
2)    qPCR RdRp gene: in figure 2G-H, line 316-329. Please indicated the copy-number/mL of the RdRp gene detected or use estimated PFU/mL similar to Fig A2D.
3)    The sample size of each experiment was limited, did all experiments been repeat duplicated?
4) The effect of resveratrol and pterostilbene on the primary human cells were not been described in detail. The author should describe any detrimental effects including the change of TEER or the cell morphology observed during the experiments while the viability of the ALI cultures was not a major indicator for the cell damages. 
5) Did these two chemicals induce interferon or other innate immunity? The direct effect of interferon stimulation may largely influence the outcome of virus infection on ALI cultures. Please include it in the discussion. 

Author Response

See attachment

This manuscript is a resubmission of an earlier submission. The following is a list of the peer review reports and author responses from that submission.

Round 1

Reviewer 1 Report

General Comments:

This is an interesting paper and appears scientifically sound.

In terms of format the paper would benefit from moving some of the controls from the appendix into the main body of the text as questions such as, are the compounds being screened for antiviral activity toxic are and is any toxicity impacting upon PFU number are both obvious and also answered by the authors.

There is a small tendency to generalise and overinterpret. This is evident on line 61 where mechanisms are not cited and again in the discussion. In the discussion it is assumed that the reduction in pfus is a function of the drug inhibiting viral reproduction. In a general sense this is not incorrect however there is not specific mechanistic data to support this. Consequently it may be a good idea to tone this extrapolation down a little.

Specific comments:

Line 61. How? What s the mechanism? Same comments on line 63-5

Line 78. Best not to start sentences with numbers.

Line 78. Where are Bulkpowders based ?

Line 83: % (w/v) or (v/v) and through out i.e. line 86 and 88 and others

Line 96: define ALI

Reviewer 2 Report

In this study, the authors evaluate the antiviral properties of resveratrol and its metabolically more stable analog, pterostilbene, against SARS-CoV-2 in three different cell culture infection models. Both resveratrol and pterostilbene are reported to have activity against a range of DNA and RNA viruses, although it is unclear if there is a common mode of action against these different viruses. Although not cited, an earlier study by Yang et al. 2020 (DOI: 10.1002/ptr.6916) also reported that resveratrol can inhibit SARS-CoV-2 replication in Vero cells.

Data is presented to show that neither compound inactivates the infectious particles themselves (i.e. they are not virucidal) and instead act within the first four hours or less of the infection cycle. This begs the question of where exactly in the infection/replication process the production of new infectious virus is being impacted. The chief strength of this study is use of an air-liquid interface primary human bronchial epithelia model. The replication kinetics are slower compared to Vero cells but drug treatment yields are stronger inhibitory effect.

What is missing in this and similar studies of resveratrol and pterostilbene is a clear understanding of how the compounds exert their antiviral effect. While the activity of these compounds against SARS-CoV-2 is timely, the lack of information on how the virus is impacted is a weakness that could be addressed without too much difficulty. For example, amplification of the incoming RNA genome and synthesis of the sub-genomic RNAs by the viral RNA polymerase can be monitored by RTqPCR. Likewise, commercial antibodies to several viral proteins are now available. A few additional experiments along these lines could potentially answer the basic questions of whether viral gene expression is impacted or whether the compounds act on later events such as the assembly of new particles.

MINOR SUGGESTIONS

Both compounds appear to be less effective in Calu3 cells another transformed cell line that is permissive for SARS-CoV-2. Unfortunately, the Calu3 analysis is limited by greater impacts on cell viability thereby limiting the concentrations that could be meaningfully tested. As such, it is unclear what useful conclusion this data provides.

Appendix A Fig 2. The interpretation of these time-of-addition data is unclear as none of the effects are statistically significant. 

Reviewer 3 Report

The manuscript describes the antiviral activity of resveratrol and pterostilbene against SARS-CoV-2 in vitro models.

Overall, the experimental designs were well-planed. However, the antiviral activities of the compounds were moderate at most. The antiviral activities were evident only at > 100 µM for resveratrol or > 50 µM for pterstibene, indicating a potential off-target effects to the host cells. This fact may be consistent with a broad-spectrum antiviral activity against other viruses without a clear mechanism of action.

  1. Title : the antiviral activities demonstrated in the manuscript was moderate. The word of “Potently” should be revised accordingly.

  1. Cell toxicity assays : The assays used in the paper were cell viability assays ,not cytotoxicity assay. Change in cell viability by compounds require a longer period of exposure. However, the paper used a 8 hours period for the cell viability test. A longer exposure such as three days of exposure is needed to determine the actual effect on cell viability. Or cytotoxicity tests ( e.g, G6PD Release Assay) with a minimum of 24-hours exposure would be needed.

2.1 Cellular cytotoxicity of resveratrol and pterostilbene in PBECs. : The data showed only 50% cell viability in the model. Is this expected and well-known fact? Needs a validation of the method or protocol.

  1. Ethanol itself seems to affect virus stability or infectivity (Figure 2). It is not clear if the concentration of ethanol was kept consistent ( 0.2 or 0.6%) for test wells with a lower concentration of resveratrol and pterostilbene to avoid any effects by variation in ethanol amount in the well.

  1. Fig 2.A needs a statistical evaluation.
  2. Fig 2. C and D : Another explanation for the figure is that this data clearly demonstrates that treatment of the cell (or virus) during the infection period affects the yield of progeny virus. This effect gets bigger due to the longer exposure during the “post” period.

  1. Antiviral effect of the two compounds in the ALI model does not correlate with data from the Vero E6 model. In Vero E6 cells, pterostilbene showed a higher antiviral activity than resveratrol; however, the ALI model showed the opposite. Explanations on this discrepancy are needed in the discussion section.

  1. Authors discussed that this high concentration can not be achieved in clinical settings. If not, more description of the implication of the study would be needed, such as modification of the compounds etc.

  1. No mechanism of antiviral activity has been described.

Reviewer 4 Report

In the manuscript by Bram M. ter Ellen et al., the authors demonstrated the efficacy of resveratrol and pterostilbene on suppression of SARS-CoV-2 replication in vitro. Moreover, they confirmed the antiviral efficacy of the compound in air-liquid interface cultured human primary bronchial epithelial cells. The topic is of interest, nevertheless, there are some limitations in the manuscript. Overall, the manuscript is written well. However, the study focuses mainly on the antiviral efficacy of compounds against SARS-CoV-2. Some comments are outlined below:

- Data should be represented either as mean ± SD if presenting scatter plot with a bar or mean ± SEM in case of simple bar blot.

- Fig. 1A: Resveratrol 120 µM is not significant? Similar comment for Fig 3B: Pterostilbene 60 µM.

- In the manuscript, only a student t-test is used to determine the statistical significance that leads the reader to a critical vision of the statistics. No multiple comparison analysis was performed despite more than two experimental groups existed. Moreover, no need to write about the statistical differences and p-values in the figure legend when there is no statistical significance in the figure.

- What is the rationale for using different MOI for antiviral and durability assays?

- Both Resveratrol and Pterostilbene have been widely studied for anti-inflammatory activity. Therefore, it is interesting to see the inhibition of SARS-CoV-2-induced cytokine and chemokine expression by these compounds.